# Causal associations between cardiorespiratory fitness and type 2 diabetes

Lina Cai[1], Tomas Gonzales [1], Eleanor Wheeler [1], Nicola D. Kerrison[1], Felix R. Day [1], Claudia Langenberg [1], John R. B. Perry [1], Soren Brage [1,2] & Nicholas J. Wareham [1,2] ✉

Higher cardiorespiratory fitness is associated with lower risk of type 2 diabetes. However, the causality of this relationship and the biological mechanisms that underlie it are unclear. Here, we examine genetic determinants of cardiorespiratory fitness in 450k European-ancestry individuals in UK Biobank, by leveraging the genetic overlap between fitness measured by an exercise test and resting heart rate. We identified 160 fitness-associated loci which we validated in an independent cohort, the Fenland study. Gene-based analyses prioritised candidate genes, such as *CACNA1C, SCN10A, MYH11* and *MYH6*, that are enriched in biological processes related to cardiac muscle development and muscle contractility. In a Mendelian Randomisation framework, we demonstrate that higher genetically predicted fitness is causally associated with lower risk of type 2 diabetes independent of adiposity. Integration with proteomic data identified N-terminal pro B-type natriuretic peptide, hepatocyte growth factor-like protein and sex hormone-binding globulin as potential mediators of this relationship. Collectively, our findings provide insights into the biological mechanisms underpinning cardiorespiratory fitness and highlight the importance of improving fitness for diabetes prevention.

Cardiorespiratory fitness, which we refer to in this paper as fitness, is the ability of the circulatory and respiratory systems to supply oxygen to working muscles during prolonged exercise[1,2]. Fitness is a multifactorial trait[2]. Twin and family studies suggest that fitness has a strong genetic component, with an estimated heritability of 40–70%[3–6]. Fitness can be improved through exercise training[3,7,8]; however, it is important to recognise that fitness is a complex trait, distinct from physical activity behaviour, representing a dimension of physical health that independently predicts various health outcomes[9–13].

Higher fitness is associated with higher insulin sensitivity independently of the effect of physical activity, which in turn, could reduce the risk of cardiometabolic diseases including type 2 diabetes[14,15]. A recent meta-analysis of 22 observational studies incorporating 1.6 million individuals (40,286 incident type 2 diabetes cases) reported that the relative risk of type 2 diabetes was 8% lower per 1 metabolic equivalent of task (MET) increment in fitness, after adjusting for adiposity[16]. However, improving fitness has not been an explicit intervention target in any of the large diabetes prevention trials that have been conducted to date; thus, it remains unclear whether fitness is causally linked to type 2 diabetes risk. Using a genetic risk score for fitness as an instrumental variable in Mendelian randomisation analyses offers a way of evaluating whether fitness is causally linked to type 2 diabetes as this approach reduces the risk of confounding and reverse causality which can affect observational studies[17].

The HERITAGE Family Study identified several loci that may be associated with fitness[18]. However, this study was undertaken in a relatively small cohort with limited statistical power and moderate coverage of the genome. More recently, an analysis of UK Biobank data identified genetic loci associated with fitness which are enriched among genes associated with various cardiometabolic diseases[19]. However, interpretation of these results is problematic, because fitness estimates from the risk-stratified submaximal bike test were

[1]MRC Epidemiology Unit, University of Cambridge, Cambridge, UK. [2]These authors contributed equally: Soren Brage, Nicholas J Wareham.
✉e-mail: nick.wareham@mrc-epid.cam.ac.uk

derived using a method recently shown to be biased for UK Biobank bike test data[13]. No studies have employed validated genetic instruments for fitness to investigate causality of the association with disease outcomes.

Here, we conduct a genome-wide association study of validated fitness estimates from the subsample with bike test data in the UK Biobank cohort and triangulate the results with a genome-wide association study of resting heart rate in the full cohort to develop an optimised genetic instrument for fitness. We then validate our instrumental variable in an independent cohort (Fig. 1). We apply this instrument to evaluate the causal relationships between fitness and diabetes risk, as well as intermediate risk traits using Mendelian randomisation methods. Finally, we explore the associations between genetically predicted fitness and blood protein levels and conduct bioinformatic analyses to gain further insights into potential biological pathways.

## Results

### Observational association between fitness and diabetes
We defined fitness as maximal oxygen consumption ($VO_2max$, expressed in ml $O_2$ per min per kg fat-free mass (FFM) in this study) estimated from a submaximal ramped cycle ergometer test[20] in the UK Biobank study. The validation of these fitness estimates and a description of the baseline characteristics of participants are reported elsewhere[13]. In general, fitter people were younger, more likely to be male, more physically active, had higher education level, better lung function, lower body mass index (BMI) and lower resting heart rate. The mean (±standard deviation) fitness level for the 34,179 men included in this analysis was 43.1 (±6.4) ml $O_2 \cdot min^{-1} \cdot kg^{-1}$ FFM, and it was 39.8 (±7.1) ml $O_2 \cdot min^{-1} \cdot kg^{-1}$ FFM for the 39,395 women (Supplementary Table 1). There were 1,852 cases of incident type 2 diabetes in the total sample of 73,574 people with fitness measurements during 10 years of follow-up. Higher fitness was strongly inversely and linearly associated with risk of developing type 2 diabetes (Fig. 2). Each 1 ml $O_2 \cdot min^{-1} \cdot kg^{-1}$ FFM higher fitness was associated with 3% (95% Confidence Interval (CI): 2-4%) lower risk of developing type 2 diabetes after adjustment for age, sex, adiposity and other potential confounders (Supplementary Table 2); this equates to 19% lower risk for every standard deviation higher fitness.

### Genome-wide association study of fitness
After excluding individuals of non-European genetic ancestry and those without available genotype data, we performed a genome-wide association study (GWAS) on fitness in 69,416 participants of European Ancestry in the UK Biobank study. We identified 14 genome-wide significant fitness-associated SNPs (Supplementary Table 3); the Manhattan plot and the Q-Q plot of the GWAS are shown in Supplementary Fig. 1. The genomic inflation factor $\lambda_{GC}$ was 1.15 and the Linkage Disequilibrium (LD) score regression intercept was 1.02 (s.e. = 0.01), which suggests that the slight inflation observed was primarily attributed to the polygenic nature of the fitness trait. A genome-wide single nucleotide polymorphism (SNP)-based heritability of 12.7% (s.e. = 1.0%) was estimated using LD score regression[21].

### Cardiorespiratory fitness and resting heart rate
Resting heart rate is inversely correlated with fitness in observational studies and decreases as a response to aerobic exercise training[2,22–25]. We hypothesised that resting heart rate could also be used as a viable proxy trait for fitness in a genetic framework and conducted several analyses to test this. In UK Biobank, the resting heart rate measure is available in nearly all participants. We leveraged the genetic overlap between fitness and resting heart rate and the large sample size for resting heart rate to optimise the genetic prediction of fitness, accounting for resting heart rate associated variants that are not related to fitness.

To do this, we conducted a GWAS of resting heart rate among >450k UK Biobank participants of European Ancestry and identified 427 distinct genome-wide significant variants ($p < 5 \times 10^{-8}$) (see Methods). We found a strong inverse genetic correlation between fitness and resting heart rate (Rg = −0.68, s.e. = 0.03, $p = 5.4 \times 10^{-120}$) using LD score regression[21]. In secondary analyses, we found that the genetic correlations between fitness and other physiologically relevant

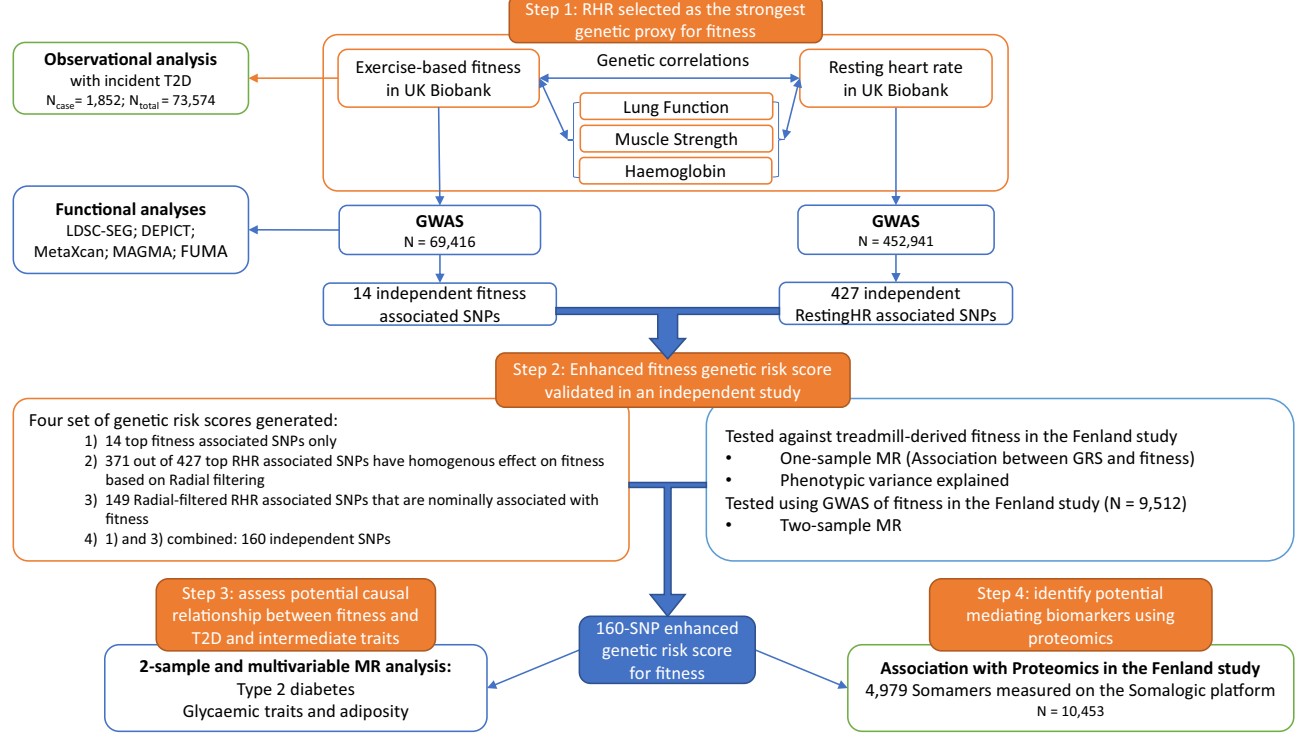

**Fig. 1 | Flow chart of the study.** The figure shows the different steps in describing the association between cardiorespiratory fitness and incident type 2 diabetes, the derivation of a genetic score for fitness for the Mendelian randomisation analysis and the investigation of mediation using proteomics.

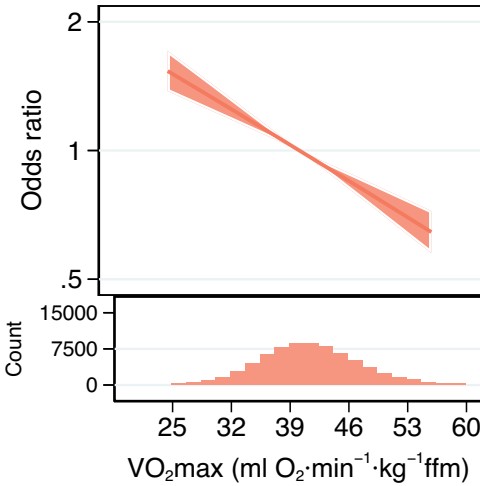

**Fig. 2 | Observational association between fitness and risk of type 2 diabetes.** Odds Ratio (and 95% confidence interval) for the association of cardiorespiratory fitness level with incident type 2 diabetes after adjustment for potential confounding variables (Model 3, Supplementary Table 2) and when using 41 ml $O_2 \cdot min^{-1} \cdot kg^{-1}$ fat-free mass (FFM) as the reference level. Histogram bins are 1.75 ml $O_2 \cdot min^{-1} \cdot kg^{-1}$ FFM wide and represent the distribution of cardiorespiratory fitness levels in UK Biobank participants ($n = 73,574$) who completed exercise testing. Source data are provided as a Source Data file.

traits (lung function, handgrip strength and haemoglobin) were not as strong as the correlation between fitness and resting heart rate (Supplementary Table 4). The genetic correlations between resting heart rate and these traits were comparable with those for fitness (Supplementary Fig. 2).

Bi-directional Mendelian Randomisation analysis between resting heart rate and fitness was conducted. The result was compatible with a bi-directional causal relationship ($p < 0.05$) (Supplementary Fig. 3). We compared the effect estimates of the 14 genome-wide significant fitness-associated SNPs and their look-ups from the resting heart rate GWAS results and vice versa (comparing the effects of 427 independent resting heart rate-associated SNPs with those in the fitness GWAS). Although the two traits are strongly genetically correlated, the results show that there are some genetic determinants of resting heart rate that are independent of fitness. Therefore, further prioritisation and validation of genetic instruments for fitness is required.

### Prioritisation and validation of genetic instruments for fitness
We developed a statistical framework to generate a robust genetic instrument for fitness by triangulating the genetic basis of fitness and resting heart rate. We validated this optimised genetic instrument in an independent study before taking it forward for subsequent Mendelian Randomisation (MR) and proteomics analyses.

To optimise the genetic instrument for fitness, we constructed four instruments for fitness using various criteria and evaluated the validity and strength of these instruments with treadmill-measured fitness in an independent cohort, the Fenland study (see Methods). The genetic instruments were calculated as the sum of the number of fitness-increasing alleles at each locus carried by each individual, weighted by the corresponding effect size from the associations with fitness in UK Biobank. The radial-filtering approach (see Methods) identified 148 resting heart rate variants which showed a consistent effect with fitness. We combined the 14 variants from the exercise test-based GWAS with 146 of these 148 variants, representing additional loci, resulting in a fourth instrument of 160 variants.

Each of the four instruments was significantly associated with treadmill-measured fitness in the Fenland study (Supplementary Table 5). The instrument with 14 distinct genome-wide significant

fitness-associated variants explained 0.54% of the phenotypic variance in fitness in Fenland but the instrument with 160 genetic variants was the strongest of the four, explaining 1.08% of the variance. We therefore selected this 160-variant instrument as the optimal instrumental variable for fitness to be taken forward in further analyses (Supplementary Data 1). The difference in mean fitness levels between participants in the top and bottom decile of this optimised fitness instrument was 3.6 ml $O_2 \cdot min^{-1} \cdot kg^{-1}$ FFM (Supplementary Fig. 4). We also examined the association between this genetic risk score of fitness and device-measured physical activity in the subsample of white unrelated Europeans with accelerometry data in UK Biobank ($n = 71k$); we found a positive correlation for both overall physical activity volume ($p = 1.1 \times 10^{-4}$) and time spent in at least moderate intensity activity ($p = 3.7 \times 10^{-4}$).

### Evaluating the impact of fitness on risk of type 2 diabetes
Using the 160-SNP instrumental variable for fitness (157 SNPs including proxies with both fitness and type 2 diabetes GWAS results), we performed a two-sample inverse variance weighted (IVW) MR analysis[26] to assess the potential causal effect of fitness on risk of developing type 2 diabetes (Table 1). We observed a nominally significant association between higher fitness and lower risk of type 2 diabetes (OR = 0.97 per 1-unit higher genetically predicted fitness in ml $O_2 \cdot min^{-1} \cdot kg^{-1}$ FFM, 95% CI: 0.94–1.00; $p = 0.086$); however, there was evidence of significant heterogeneity (Cochran's Q $p$-value < 0.001), potentially indicating horizontal pleiotropy (Supplementary Table 6). Therefore, we removed variants identified as outliers using the Radial method[27] (see Methods) resulting in a filtered score containing 126 variants, for which we observed a significant and directionally consistent causal association between fitness and type 2 diabetes; 1-SD higher genetically predicted fitness was associated with an 11% (95% CI: 4-18%; $p = 0.005$) lower risk of developing type 2 diabetes (Fig. 3), with no evidence of heterogeneity (Cochran's Q $p$-value = 0.686). A series of sensitivity analyses were conducted such as MR Egger[28] and weighted median models[29]. The result was consistent using the MR-PRESSO method[30] and also supported by one-sample MR analysis using the optimised genetic instrument as a proxy for fitness in an analysis of association with prevalent type 2 diabetes in UK Biobank (Supplementary Table 6).

### Evaluating the impact of fitness on intermediate metabolic traits
We also assessed whether genetically predicted fitness was causally associated with glycaemic traits (fasting insulin, fasting glucose, 2-h post-75 g oral glucose load glucose (2-h glucose) and HbA1c) or adiposity (BMI). Similar analytical strategies were applied to each of the intermediate traits as for the MR analyses with type 2 diabetes. We observed a significant association of genetically predicted fitness with fasting insulin after Bonferroni correction for multiple testing ($p = 0.001$), but not with other traits ($p > 0.01$). (Table 1: Inverse variance weighted results; Supplementary Table 7: all results).

We conducted multivariable MR to assess the direct effect of genetically predicted fitness on type 2 diabetes risk, after adjusting for the effects of the same set of fitness-associated variants on glycaemic traits or adiposity. We found that the casual association between fitness and type 2 diabetes that was observed in the univariate Radial-filtered MR was attenuated but remained nominally significant ($p < 0.05$) after controlling for the effects of intermediate traits, individually and collectively (Fig. 3; Supplementary Table 8).

### Association with proteins
We examined the association between genetically predicted fitness levels and circulating protein levels of 4775 protein targets assessed by the aptamer-based technology (SomaScan©) in 10,707 individuals from the Fenland study. We observed significant associations between genetically predicted fitness and higher N-terminal pro B-type

**Table 1 | Two-sample Mendelian randomisation results using inverse-variance weighted analyses of genetically predicted cardiorespiratory fitness on type 2 diabetes and related intermediate traits**

| Outcome | Radial-filtered | n_SNPs | beta | s.e. | p | CochQp | EGGER intercept p-value |
|---|---|---|---|---|---|---|---|
| Type 2 diabetes[a] | No | 157 | −0.0284 | 0.0165 | 0.086 | <0.001 | 0.330 |
| | Yes | 126 | −0.0171 | 0.0060 | **0.005** | 0.69 | 0.780 |
| fasting insulin | No | 156 | −0.0073 | 0.0052 | 0.160 | <0.001 | 0.543 |
| | Yes | 134 | −0.0112 | 0.0032 | **0.001** | 0.79 | 0.562 |
| fasting glucose | No | 156 | 0.0011 | 0.0057 | 0.854 | <0.001 | 0.702 |
| | Yes | 140 | −0.0002 | 0.0027 | 0.926 | 0.98 | 0.423 |
| 2-hr glucose | No | 156 | −0.0157 | 0.0072 | 0.029 | <0.001 | 0.152 |
| | Yes | 141 | −0.0099 | 0.0051 | 0.052 | 0.86 | 0.595 |
| HbA1c | No | 155 | −0.0011 | 0.0048 | 0.821 | <0.001 | 0.426 |
| | Yes | 137 | −0.0009 | 0.0031 | 0.775 | 0.74 | 0.325 |
| BMI | No | 157 | −0.0033 | 0.0038 | 0.388 | 0.00 | 0.778 |
| | Yes | 108 | −0.0024 | 0.0015 | 0.103 | 0.11 | 0.470 |

[a] The effect sizes (betas) for type 2 diabetes MR analyses are logOdds.

2-hr glucose = 2-h post-load plasma glucose, HbA1c = glycated haemoglobin, BMI = BMI.

The effect estimates of the optimised genetic instruments on type 2 diabetes were extracted from the meta-analysis of GWAS summary statistics from the DIAMANTE consortium excluding UK Biobank participants (55,005 cases, 400,308 controls)[55]. The effect estimates on FI, FPG, 2hrPG and HbA1c were extracted from meta-analysis summary statistics for these traits among European population acquired from the MAGIC investigators working group[56]. The effect estimates on BMI were obtained from the publicly available meta-analysis results of BMI GWAS from Genetic Investigation of Anthropometric Traits (GIANT) consortium and UK Biobank[57].

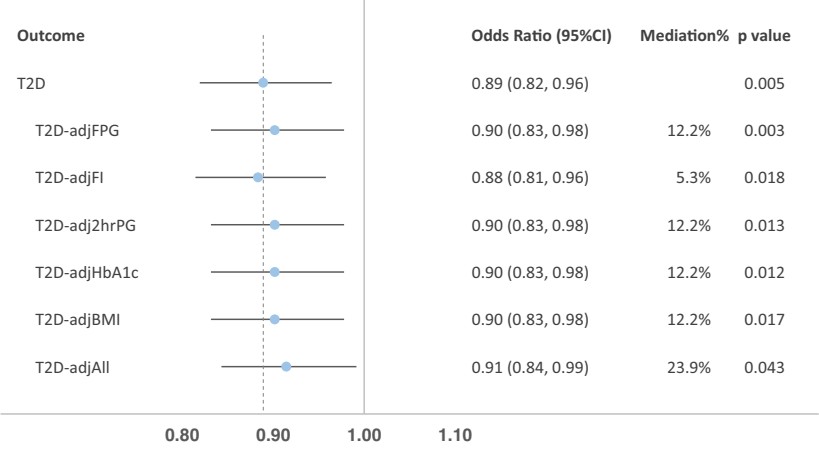

| Outcome | | Odds Ratio (95%CI) | Mediation% | p value |
|---|---|---|---|---|
| T2D | | 0.89 (0.82, 0.96) | | 0.005 |
| T2D-adjFPG | | 0.90 (0.83, 0.98) | 12.2% | 0.003 |
| T2D-adjFI | | 0.88 (0.81, 0.96) | 5.3% | 0.018 |
| T2D-adj2hrPG | | 0.90 (0.83, 0.98) | 12.2% | 0.013 |
| T2D-adjHbA1c | | 0.90 (0.83, 0.98) | 12.2% | 0.012 |
| T2D-adjBMI | | 0.90 (0.83, 0.98) | 12.2% | 0.017 |
| T2D-adjAll | | 0.91 (0.84, 0.99) | 23.9% | 0.043 |

Odds Ratio of T2D per 1-SD increase of CRF (ml O²/min/kg FFM)

**Fig. 3 | Forest plot for Mendelian randomisation analysis results for the genetically predicted effect of cardiorespiratory fitness on type 2 diabetes, and multivariable Mendelian randomisation analyses of cardiorespiratory fitness on type 2 diabetes after adjustment for the effects of glycaemic traits and BMI.** Odds Ratios presented are based on Inverse Variance Weighted MR after using Radial-filtered instrumental variable. Mediation % represents the proportion of effect of fitness on T2D that is mediated by the intermediate traits, i.e. glycaemic traits and BMI. SD: standard deviation; CRF: cardiorespiratory fitness; adjFPG: adjusted for fasting plasma glucose; adjFI, adjusted for fasting insulin; adj2hrPG, adjusted for 2-h plasma glucose after oral glucose tolerance test; adjHbA1c, adjusted for HbA1c; adjBMI, adjusted for BMI; adjAll, adjusted for all the intermediate traits above. Sample sizes for each individual MR analysis are provided as a Source Data file.

natriuretic peptide (beta per 1-SD higher fitness = 0.058, s.e. = 0.008, $p = 9.45 \times 10^{-13}$) and lower hepatocyte growth factor-like protein (beta = −0.042, s.e. = 0.009, $p = 2.59 \times 10^{-6}$) after Bonferroni correction for the number of aptamers tested ($0.05/4,979 = 1.004 \times 10^{-5}$) (Fig. 4; Supplementary Table 9). Although not reaching Bonferroni-corrected significance, we observed a nominal association between genetically predicted levels of fitness and higher sex hormone-binding globulin (beta = 0.026, s.e. = 0.008, $p < 0.001$). Studies have shown that higher levels of sex hormone-binding globulin reduce the risk of type 2 diabetes[31], suggesting a potential mechanism mediating the lower risk of type 2 diabetes with increased levels of fitness. Further MR analysis in the UK Biobank study provided supporting evidence that genetically predicted higher fitness is associated with higher levels of sex hormone-binding globulin ($p = 9.6 \times 10^{-11}$) (Supplementary Table 7).

Multivariable MR analysis results suggested that sex hormone-binding globulin attenuated the effect of fitness on type 2 diabetes but not completely (Supplementary Table 8).

### Gene-set enrichment and biologically relevant tissue and cell types

To better understand biology, we used DEPICT[32] to prioritise gene-sets and biological pathways that were enriched with fitness-associated genes. Several top gene-sets were linked to biological processes including muscle cell differentiation, muscle tissue and organ development, as well as increased cardiac muscle contractility (FDR < 0.20) (Supplementary Table 10). These findings were also supported by results obtained from MAGMA (Supplementary Table 11) and from the FUMA 'GENE2FUNC' pipeline (Supplementary Table 12).

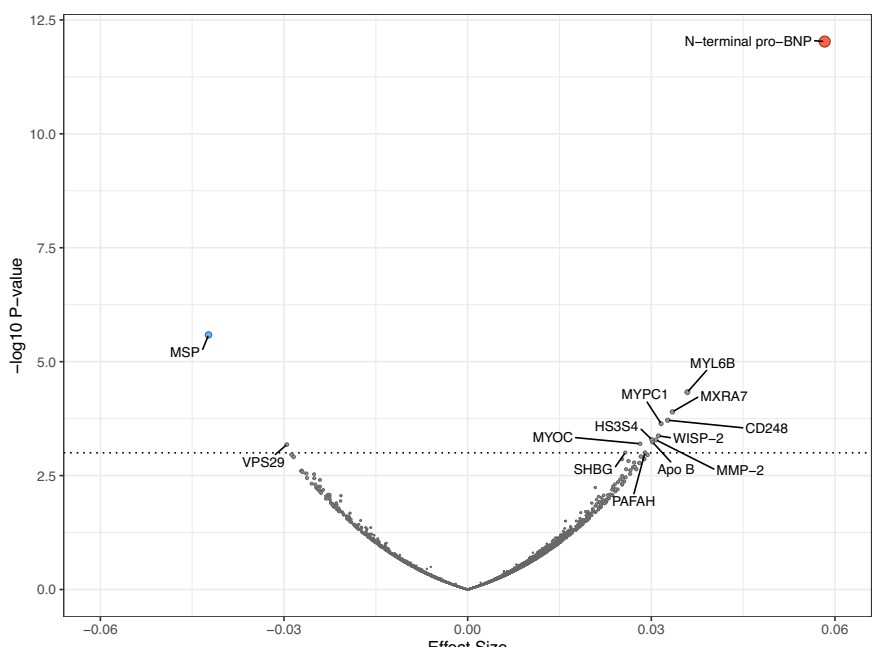

**Fig. 4 | Volcano plot of associations between the enhanced 160-SNP genetic risk score for cardiorespiratory fitness and Protein Targets assessed by the aptamer-based technology (SomaScan©) in 10,707 individuals from the Fenland study.** Each dot represents a Somamer targeting a protein. The horizontal dashed line represents $p < 0.0001$ and Somamers with $p < 0.0001$ are annotated. N-terminal pro-BNP, N-terminal pro B-type natriuretic peptide, MSP, Hepatocyte growth factor-like protein, MYL6B Myosin light chain 6B MXRA7 Matrix- remodelling-associated protein 7, CD248 Endosialin, MYPC1 Myosin-binding protein C, slow-type, WISP-2, WNT1-inducible-signalling pathway protein 2, HS3S4 Heparan sulfate glucosamine 3-O-sulfotransferase 4, MMP-2 72 kDa type IV collagenase, Apo B Apolipoprotein B, MYOC Myocilin, VPS29 Vacuolar protein sorting-associated protein 29, SHBG Sex hormone-binding globulin, PAFAH Platelet-activating factor acetylhydrolase. Source data are provided as a Source Data file.

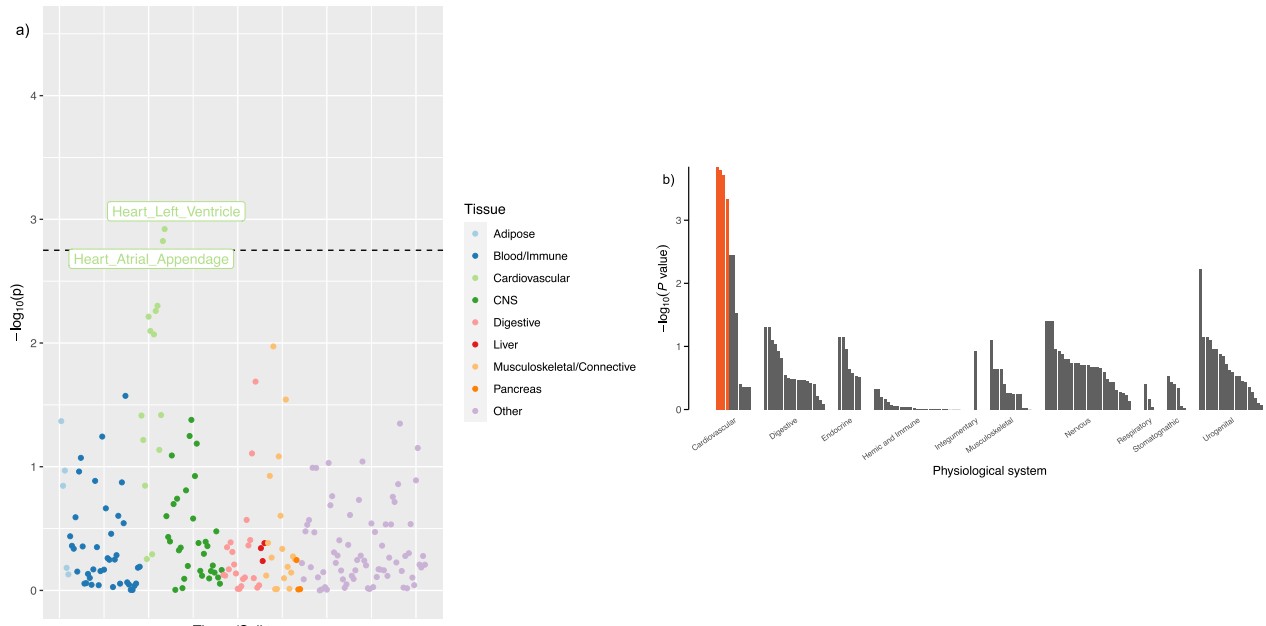

**Fig. 5 | Tissue and cell-type specific enrichment analysis for cardiorespiratory fitness using LDSC-SEG and DEPICT. a** Expression enrichment in a total 207 tissues and cell type specific expression data from GTEx v7 and Franke lab. Each dot represents a tissue or cell type further categorised into 9 general tissue groups indicated by different colours. **b** Results from DEPICT using independent fitness-associated variants ($p < 10^{-5}$). Each bar represents a tissue or cell which is categorised in 10 physiological systems. Orange bar marks the four significantly enriched tissues (FDR < 0.05).

To identify tissues that are more likely involved in the biological functions of fitness, we used LDSC-SEG[33] to identify tissue and cell type-specific enrichment using the fitness GWAS summary statistics. We observed that the expressions of fitness-associated loci were enriched in heart tissues (Fig. 5a; Supplementary Figs. 6 and 7).

Consistently, significant enrichment in four cardiac tissues and cell types that belong to the cardiovascular system, including the heart, atria, atrial appendage, and ventricular tissues were found using DEPICT with input of independent genetic variants associated with fitness at a suggestive significance level ($p < 10^{-5}$) (Fig. 5b;

Supplementary Table 12). In addition, directional tissue specificity profiling in FUMA also indicated that fitness-associated genes were enriched in up-regulated differentially expressed genes sets in heart tissue (Bonferroni corrected $p < 0.05$) (Supplementary Fig. 8).

## Discussion

In this large cohort study, we observed a strong linear inverse association between exercise-measured cardiorespiratory fitness and the risk of developing type 2 diabetes. We used the availability of fitness and resting heart rate data at scale to develop an enhanced 160-SNP genetic risk score for fitness, which we validated in an independent study. Using this genetic risk score in a Mendelian randomisation analysis, we observed an association between fitness and type 2 diabetes risk that was compatible with an underlying causal relationship. Analyses with intermediate traits showed a probable causal relationship with fasting insulin, a measure of insulin resistance known to be linked to fitness. Novel potential intermediate pathways between fitness and type 2 diabetes were also highlighted by the association between genetically predicted fitness and N-terminal pro B-type natriuretic peptide, hepatocyte growth factor-like protein and sex hormone binding globulin.

Cardiorespiratory fitness is usually defined by VO2max, the maximum oxygen uptake during high-intensity exercise, which is considered the best indicator of cardiorespiratory capacity. There are safety concerns to measuring fitness using a maximal exercise test in large-scale epidemiological studies. Although some studies have used extensive pre-testing selection criteria to exclude individuals in whom the test might be risky, this results in tests that are undertaken only in a population of relatively healthy fit individuals, with obvious selection biases and lack of generalisability. Therefore, submaximal tests have been developed, like the bike ergometer test deployed in the UK Biobank study which allows safe measurement in the greatest number of individuals, thus reducing selection bias whilst preserving validity, as indicated by the strong relationship with the gold standard measurement in validation studies[13]. Additionally, we used the Fenland study to validate and prioritise the genetic instruments constructed in the UK Biobank. There are differences in the tests used to measure fitness between the two studies[13,34], principally that a treadmill test was used in the Fenland study which is a weight-bearing exercise modality. Nonetheless, the successful validation of the derived fitness instrument using data from a different study design may also be a strength since correlation of errors and biases intrinsic to one test modality is less likely to explain strong validation results.

In this study, we observed a strong inverse genetic correlation between resting heart rate and fitness in UK Biobank, and given the large sample size available for resting heart rate, we used the overlapping genetic basis of resting heart rate and fitness to create a more robust genetic instrument for fitness than would have been possible using only exercise-based fitness estimates alone. We applied the Radial method to prioritise resting heart rate-associated variants that were associated with fitness, which were then combined with the top fitness-associated variants to create an enhanced genetic instrument for fitness. The proportion of variance explained by this enhanced instrument in the phenotypic fitness level in the independent dataset was double that of the instrument made from only the 14 genome-wide significant variants identified in the analysis of fitness.

When applying the instrument to examine the association with type 2 diabetes risk in MR analysis, we observed an association that was compatible with a causal role for fitness. In a recent meta-analysis among 40,286 incident cases of type 2 diabetes and a total of 1,601,490 participants, each 3.5 ml O2·min⁻¹·kg⁻¹ body mass or 1 metabolic equivalent (MET) higher fitness was associated with an 8% (95% CI: 6%-10%) lower relative risk of type 2 diabetes[16]. In the UK Biobank study, we observed 3% lower risk of diabetes per 1 ml O2·min⁻¹·kg⁻¹ fat-free mass, equivalent to 19% lower risk per standard deviation, using the fat-free mass scaled fitness measure and adjusting for fat mass. To put this in a comparable scale as the previous study, we observed a 13% lower risk of diabetes per 1 MET higher fitness adjusting for BMI. Based on 2-sample MR analyses, we observed 11% lower risk of diabetes per standard deviation higher genetically predicted fitness.

A further investigation of intermediate traits for type 2 diabetes suggested that genetically predicted higher fitness was also significantly associated with lower fasting insulin, a marker of insulin resistance. The association between fitness and insulin sensitivity is biologically plausible based on their common underlying physiological function linked with the oxidative capacity of the skeletal muscle[14,35,36]. However, the direct effect of fitness on type 2 diabetes risk was only slightly attenuated but still nominally significant in the multivariable MR model, which suggests that the association between fitness and fasting insulin did not fully explain the causal effect of fitness on type 2 diabetes risk and that other mechanisms may be involved.

In contrast with fasting insulin, we did not find convincing evidence supporting a causal relationship between fitness and other glycaemic parameters, although the association with the 2-h glucose levels was close to nominal significance. The effect size was comparable to that for fasting insulin but may not have been statistically significant because the GWAS cohort for the 2-h glucose level is so much smaller than for other traits. In the multivariable MR analyses, adjustment for BMI did not attenuate the apparent causal effect of fitness on type 2 diabetes risk, which is concordant with the apparent BMI-independent observational association of fitness on type 2 diabetes risk observed both here and in a recent large meta-analysis of prospective cohort studies[16].

We explored other potential pathways that might mediate the association of fitness on type 2 diabetes risk, given the observed strong link between fitness and fasting insulin levels did not wholly explain the causal association between fitness and diabetes. In an analysis capitalising on the availability of proteomics alongside fitness measurements in the Fenland study, we observed a potential mediating role of N-terminal pro B-type natriuretic peptide, which is compatible with reports of a previous MR analysis which suggests that one standard deviation genetically predicted higher N-terminal pro B-type natriuretic peptide levels was associated with a 21% reduction in type 2 diabetes risk[37]. The observation in this study of a possible mediating role for sex hormone binding globulin is also supported by strong genetic evidence between sex hormone binding globulin and type 2 diabetes in previous reports[31,38]. The results point to novel pathways beyond glycaemia and insulin resistance which may link fitness and type 2 diabetes and are relevant not only to future studies of the biological basis of fitness but would also be of relevance as potential targets for therapies that mimic the metabolic health benefits of improved fitness.

A previous genetic study of fitness in 497 sedentary individuals from the HERITAGE Family Study highlighted some genes involved in the underlying mechanisms in the cardiovascular system, skeletal muscle function, haematopoiesis, and metabolism based on bioinformatic analyses and evidence from knockout mouse models[18]. Despite the extensive search centred on biological relevance, the sample size for the original genetic discovery was small. In this study, we did not replicate any of the genes suggested by their study. Nonetheless, this study identified multiple genes that encode proteins that play key roles in cardiac and smooth muscle development and function (such as CACNA1C, SCN10A, MYH6, MYH7, MYH11), which was also supported by gene-set enrichment analyses and tissue-specific expression patterns.

This current study has certain limitations. For instance, we applied the Radial method to filter outliers in the association between resting heart rate and cardiorespiratory fitness to construct a more robust genetic instrument for fitness. Despite the notable improvement in the enhanced instrument, the causal assessment using this instrument needs to be interpreted with caution. It is possible that the association

was driven by the selected resting heart rate-associated variants, given both observational and MR studies have found a significant positive association between resting heart rate and type 2 diabetes[39,40].

In conclusion, we developed an enhanced genetic instrument for cardiorespiratory fitness by leveraging the strong genetic overlap between fitness and resting heart rate. We applied this to confirm that the observed strong association between fitness and type 2 diabetes is likely to be causal and partly mediated by the effect of fitness on insulin resistance. The study provides insights into the biological mechanisms that may explain between-individual differences in fitness and also into the pathways that might mediate the beneficial metabolic effects of higher fitness. The nature of the relationship demonstrated in this study should reignite interest in fitness as a quantifiable parameter that is causally linked to type 2 diabetes risk and which could be assessed not only in future prevention trials, but also in clinical practice in patients at risk and in public health surveillance studies, as an important determinant of metabolic health.

## Methods

### Measurement of cardiorespiratory fitness and resting heart rate in the UK Biobank study

Our primary analyses were conducted in the UK Biobank study. The study design and details of the study have been described previously (UK Biobank. https://www.ukbiobank.ac.uk/). In brief, the UK Biobank is a large-scale population-based cohort study including 503,325 participants (aged 40–69 years) recruited through the general practitioners within the UK National Health Service. Participants were enroled in 22 study centres in England, Scotland and Wales, and provided extensive data on their demographic information, medical history and health behaviour through questionnaires. Blood samples and physical measurements were taken at baseline. The study was approved by the North West Multi-Centre Research Ethics Committee. All participants provided written informed consent.

Cardiorespiratory fitness was assessed in a subsample using heart rate response to a submaximal ramped cycle ergometer test that was individualised for participant characteristics, including cardiovascular disease risk. The protocol for measuring fitness has been described in detail in the UK Biobank Cardio Assessment manual[20]. Briefly, participants were categorised into four risk groups based on a risk assessment questionnaire, where the 'minimal' and 'small' risk groups were to complete an individualised ramp test, a flat test for the 'medium' risk group, and a resting electrocardiograph (ECG) protocol for people deemed in the 'high' risk group. The test protocol was specified according to the participants' age, sex, height, weight, and resting heart rate. During the test, heart rate response was monitored and recorded using a 4-lead ECG device. All participants assigned to the bike test exercised for 6 min, followed by a 1-min motionless recovery-phase on the cycle ergometer. The derivation and validation of fitness estimates have been reported by Gonzales et al.[13]. Fitness was defined using estimated maximum oxygen consumption ($VO_2max$) values (expressed in ml $O_2$ $min^{-1}$ $kg^{-1}$ fat-free mass and, in sensitivity analyses in ml $O_2$ $min^{-1}$ $kg^{-1}$ body mass). After quality control, a total of 76,872 participants had available fitness data and 73,574 participants were included in an observational analysis of the association between fitness and incident type 2 diabetes risk, following exclusion of prevalent diabetes[41] and missing covariates. The genome-wide association analyses of exercise test-based fitness were conducted on a sub-group of 69,416 individuals after excluding individuals without genotype data and those of non-White European ancestry. In the full cohort, the automated pulse rate reading during seated blood pressure measurement was used as resting heart rate values. Participants who did not have a resting heart rate measurement ($n = 15,167$) and those who were taking beta-blockers ($n = 35,562$) for pre-existing heart conditions that could affect their heart rate were excluded. After excluding individuals without genotype data and those of non-White European

ancestry, a total of 452,941 participants were included in the analyses for resting heart rate.

### Measurement of cardiorespiratory fitness in the Fenland Study

The Fenland Study is a population-based prospective cohort study that aims to investigate the associations between genetic and environmental factors and the risk of obesity, diabetes and related metabolic traits in adults. Eligible participants were born between 1950 and 1975, and resided in Cambridgeshire and were registered at a participating General Practices in Cambridge, Ely, Wisbech and the surrounding Cambridgeshire region between 2004 and 2014. Exclusion criteria include clinically diagnosed diabetes mellitus, inability to walk unaided, terminal illness with prognosis ≤1 year at the time of recruitment, clinically diagnosed psychotic disorder, pregnancy or lactation. A total of 12,435 participants completed the baseline phenotype assessments and provided written informed consent. The study was approved by the Cambridge Local Research Ethics Committee.

Cardiorespiratory fitness was defined using estimated $VO_2max$ values (also expressed in ml $O_2$ per min per kg fat-free mass; ml $O_2$ ;min $^{-1}$ $kg^{-1}$) derived from heart rate response during a submaximal treadmill test. Participants attended one of three clinical centres at the MRC Epidemiology Unit at the University of Cambridge for a progressive treadmill test protocol consisting of three phases; a 3-min walk at 3.2 km/h followed by a 6-min increasing speed walk (phase 1), a 6-min brisk walk with an increasing gradient (phase 2) and a 4.5-min flat level jog/run phase with increasing speed up to 12.5 km/h (phase 3). The test was terminated early if the participant (1) wanted to stop, (2) reached 90% of age-predicted maximal heart rate ($208 - 0.7 \times age$)[41] or (3) had exercised above 80% of age-predicted maximal heart rate for >2 min. The treadmill protocol and derivation of the $VO_2max$ values have been described in detail elsewhere[34,42]. The procedure of $VO_2max$ estimation has been validated against directly measured $VO_2max$[43,44].

### Observational analyses

In the UK Biobank, participants self-reporting any diabetes other than gestational diabetes only, or self-reporting diabetes medication, at either touchscreen or nurse interview, or having hospital episode statistics records or death records with ICD10 codes E10-E14 and recorded or inferred diagnosis date[45] before baseline, were classified as having likely prevalent diabetes (any type); these participants were excluded from the present analysis. Participants having Hospital Episode Statistics (HES) or death records indicative of type 2 diabetes, and with an inferred diagnosis date after baseline, were classified as incident type 2 diabetes cases. Specifically, ICD10 codes used to identify likely type 2 diabetes were the presence of E11 without E10, or the presence of E14 without E10-E13.

We used logistic regression to estimate odds ratios for incident type 2 diabetes according to fitness level in UK Biobank. Logistic regression models were sequentially adjusted for age and sex (Model 1), ethnicity (White, mixed, Asian or Asian British, Black or Black British, other), hypertension (binary variable set to '1' if one of the following were observed: measured systolic blood pressure greater or equal to 140 mmHg, measured diastolic blood pressure greater or equal to 90 mmHg, or self-reported use of blood pressure medication; '0' otherwise), history of stroke, history of heart failure, history of heart disease, history of atrial fibrillation, history of chronic obstructive pulmonary disease, history of cancer, medication use (binary variables set to '1' if reported use of each of the following: beta blockers, calcium channel blockers, angiotensin-converting enzyme inhibitors, diuretics, bronchodilators, lipid-lowering agents, iron deficiency agents; '0' otherwise), smoking (never, previous, current), alcohol consumption (never, previous, current but less than three times per week, current and three or more times a week), meat intake (average consumption days per week derived from self-reported frequency of processed, beef, lamb and pork intake), oily fish intake (never, less than one per

week, one or more per week), fruit and vegetable intake (a score of '0–4' was computed from self-reported intake frequency of raw vegetables, cooked vegetables, fresh fruit, and dried fruit), salt intake (never or rarely, sometimes, usually or always), employment (unemployed, employed), and Area Deprivation Index (Model 2), and adiposity (fat mass for FFM scaled fitness, BMI for body mass scaled fitness; Model 3). Participants with prevalent type 2 diabetes at baseline were excluded from analyses.

## Genotyping and imputation

In the UK Biobank study, genotyping was performed using the UK BiLEVE and UK Biobank Axiom arrays. Initial quality control was performed by the UK Biobank with details described previously[46]. In this study, the 'v3' release of the genetic data was used, which was imputed to the full set of HRC reference panel[47] and the merged UK10K and 1000 Genomes Phase III reference panels[48]. Approximately 93 million directly genotyped and imputed autosomal genetic markers were available after quality control.

In the Fenland study, genotyping was performed using one of three genotyping arrays; the Affymetrix UK Biobank Axiom Array ($n = 8994$), the Affymetrix SNP5.0 Array ($n = 1402$) and the Illumina CoreExome-24 v1 ($n = 1060$). For the quality control of Axiom array, samples were excluded if they had failed channel contrast (DishQC <0.82), low call rate (<95%), gender mismatch between reported and genetic sex, heterozygosity outlier, unusually high number of singletons or impossible identity-by-descent values. Missing genotypes and those not directly measured were pre-phased using SHAPEIT2[49]. Genetic variants were removed if they had a call rate <95%, clusters failed Affymetrix SNPolisher standard tests and thresholds, MAF was significantly affected by plate, were a duplicate based on chromosome, position and alleles (selecting the best probe set according to Affymetrix SNPolisher), deviated from Hardy-Weinberg equilibrium ($p < 5 \times 10^{-6}$), did not match the reference or had MAF = 0[50]. The QC procedures were similar for the other arrays. Remaining variants were imputed using IMPUTE2[51] based on the HRC reference panel[47], as well as the merged UK10K and 1000 Genomes Project Phase III reference panels[48] for additional variants not available in the HRC reference panel. Post-imputation quality control of SNPs was carried out[50], and the exclusion criteria include: (1) monomorphic and singleton variants; (2) imputation quality (INFO) < 0.4 or (3) Hardy-Weinberg equilibrium $p < 5 \times 10^{-6}$. Approximately 20 million variants were available after quality control.

## Genome-wide association analyses of fitness

In the UK Biobank study, a GWAS of fitness was performed under an additive genetic model using BOLT-LMM v2.3[52] among 69,416 participants of European ancestry. Participants who did not have high-quality genotyping data, fitness or covariates values were excluded. The covariates included in the model were age at recruitment, sex, genotyping array, and the first 10 principal components to control for population structure in UK Biobank.

LD score regression (LDSC)[21] was used to assess the level of genomic inflation due to confounding bias and estimate the genome-wide SNP-based heritability using pre-calculated 1000 Genomes European LD scores provided by LDSC as the LD reference panel. All the SNPs included in the analyses were restricted to those available in HapMap Phase III to avoid confounding by variable imputation quality.

We performed distance-based clumping to identify genome-wide significant independent signals with a MAF ≥ 0.01, imputation quality score > 0.4 and at least 1 megabase (Mb) apart (assuming they would be at linkage equilibrium). We also used GCTA to perform conditional and joint analysis[53] to identify additional secondary signals using a collinearity threshold of 0.05, and selected the output variants that were at a genome-wide significance level ($p < 5 \times 10^{-8}$) both before and after conditional analyses, had <10% effect estimate change before and after conditional analyses, had MAF ≥ 0.01, and had LD correlation

<0.05. Once potential secondary signals were detected, all the selected variants at the same locus were jointly tested using a joint model to confirm their independent associations with fitness.

## Fitness genetic instruments prioritisation and validation

To leverage the shared genetic basis between cardiorespiratory fitness and resting heart rate and the large sample size available for resting heart rate, we conducted a series of analyses to derive and assess the validity and quality of genetic scores constructed by triangulating the genetic data of both traits.

First, we tested our hypothesis that resting heart rate could be a viable proxy trait for fitness in genetic settings. We conducted a GWAS of resting heart rate in the full UK Biobank cohort ($N = 452,941$ after excluding participants who were taking beta-blockers, without genotype data and of non-White European ancestry). Covariates included in the linear mixed model were age at recruitment, sex, genotyping array and first 10 principal components.

We estimated the genetic correlation between fitness and resting heart rate using LD score regression[21]. We also compared the genetic correlations between fitness and other physiologically relevant traits (lung function, handgrip strength and haemoglobin) with those with resting heart rate.

Bidirectional Mendelian Randomisation analysis between resting heart rate and fitness was conducted using previously identified, distinct genome-wide significant variants as the instrumental variable for each trait. We then compared the effect estimates of the genome-wide significant fitness-associated SNPs and their look-ups from the resting heart rate GWAS results and vice versa (comparing the effects of resting heart rate-associated SNPs with those in the fitness GWAS).

We utilised the Radial-plot method[27] (resting heart rate as exposure and fitness as outcome) to select eligible resting heart rate-associated genetic variants to construct the proxy genetic instruments for fitness by excluding heterogenous outliers. We constructed a total of four genetic instruments for fitness using various criteria. Each instrument, a weighted genetic risk score generated from each of the four sets of genetic variants, was calculated by summing the number of fitness-increasing alleles carried by each individual at each of the selected loci in the Fenland Study, weighted by its corresponding per-allele effect estimate on fitness from the GWAS summary statistics in UK Biobank. The four genetic instruments for fitness were

(Instrument 1) Genome-wide significant variants independently associated with exercise-based fitness;
(Instrument 2) Genome-wide significant resting heart rate-associated variants that passed the Radial test mentioned above and not identified as outliers;
(Instrument 3) Radial-filtered resting heart rate-associated variants that were also nominally significant in the fitness GWAS ($p < 0.05$);
(Instrument 4) Independent variants combining the list of variants from (1) and (3); variant pairs in LD ($r^2 > 0.01$) were identified and fitness variants from (1) were prioritised.

Subsequently, we evaluated the validity and strength of these instruments by comparing the significance and proportion of phenotypic variance explained in observed fitness in an independent study, the Fenland study (one-sample MR method). As a sensitivity analysis, we also conducted a fat-free mass weighted fitness GWAS in the Fenland Study. After excluding participants without fat-free mass weighted fitness phenotype and genotype data, those not of European ancestry, and related individuals, a total of 9512 individuals were included in this GWAS. QUICKTEST was used to perform GWAS among unrelated participants genotyped in each genotyping array, using an additive model adjusted for age, sex and first 5 principal components. METAL[54] was used to meta-analyse the results across each genotyping array. Next, we applied the inverse variance weighted MR method (2-sample MR) to

assess these four instruments by regressing the effect estimates of these variants from UK Biobank fitness GWAS results on the effect estimates of these variants from Fenland fitness GWAS.

Combining the results from both approaches, the genetic instrument with the strongest effect on fitness which explained the largest proportion of variance in fitness was selected and taken forward to the analyses of association with health outcomes.

## Mendelian randomisation on type 2 diabetes and related metabolic traits

We conducted two-sample Mendelian randomisation analyses to examine whether genetically predicted fitness levels were causally associated with type 2 diabetes related intermediate traits, including fasting insulin (FI), fasting plasma glucose (FPG), the 2-h post-75 g oral glucose load glucose level (2hrPG), glycated haemoglobin (HbA1c) and BMI. The effect estimates of the optimised genetic instruments on type 2 diabetes were extracted from the meta-analysis of GWAS summary statistics from the DIAMANTE consortium excluding UK Biobank participants (55,005 cases, 400,308 controls, provided by Anubha Mahajan anubha@well.ox.ac.uk)[55]. The effect estimates on FI, FPG, 2hrPG and HbA1c were extracted from meta-analysis summary statistics for these traits among European population acquired from the MAGIC investigators working group[56]. FPG, 2hrPG, HbA1c and natural-log transformed FI levels were first regressed on study-specific covariates including age, age$^2$, sex, genetic principal components and BMI (except for HbA1c) to obtain residuals, which were then rank-based inverse-normal transformed before used as the phenotypes for GWAS analyses. The effect estimates on BMI were obtained from the publicly available meta-analysis results of BMI GWAS from Genetic Investigation of Anthropometric Traits (GIANT) consortium and UK Biobank[57]. Proxies in high LD ($r^2 > 0.8$, D' > 0.8) were found for index SNPs not available in the outcome GWAS using the genotyped and imputed data of a random sample of 25,000 unrelated UK Biobank White-British participants as the reference panel, among which the common variant with the smallest $p$-value was used as the appropriate proxy. The effect alleles from the exposure and outcome GWAS were aligned to the fitness-increasing allele.

For each MR analysis, we first applied the Radial method[27] to filter out outliers, then took the remaining set of instruments forward for analyses. The inverse variance weighted method (MR-IVW)[26] with the random-effects model was used as the primary method. The Cochran's Q test[58] was used to assess heterogeneity between instruments, the MR-Egger regression[29,59] was conducted with the regression intercept examined to assess unbalanced horizontal pleiotropy, and the weighted median and penalised weighted median methods[29] which were designed to be robust with the presence of some invalid instruments were also performed. As further sensitivity analyses, Steiger filtering[60] was applied to identify reverse directionality (the association between a genetic instrument and the exposure was stronger than its association with the outcome) and MR-PRESSO[30] was also applied as an alternative approach to identify horizontal pleiotropic outliers. Subsequently, to assess the direct effect of CRF on T2D, we conducted multivariable MR to account for potential residual horizontal pleiotropic effects by adjusting for the effects of glycaemic traits or BMI in the model individually. Additionally, all intermediate traits were added to the model jointly as covariates in a full multivariable model.

MR analyses were conducted with MR-Base (http://www.mrbase.org), 'TwoSampleMR' R-package[61] or the 'MendelianRandomization' R-package developed by Yavorska and Burgess et al.[62].

## Genetic correlation with other fitness-related traits

The GWAS results for lung function traits (forced expiratory volume (FEV1), forced vital capacity (FVC), FEV1/FVC and peak expiratory flow (PEF)) were publicly available and acquired from the study by Shrine et al.[63]; the haemoglobin concentration results from Astle et al.[64].

To obtain genome-wide association results for handgrip strength, a GWAS was performed using BOLT-LMM v2.3[52] in UK Biobank. The isometric handgrip strength of UK Biobank participants was measured in an upright sitting position using a Jamar J00105 hydraulic hand dynamometer at the assessment centre at baseline[65]. The participants were asked to rest their forearms on armrests and squeeze the device as strongly as they could for about 3 s, and the maximum value reached was recorded on the device. One measurement was taken from each hand, and the higher value of the measurements from two hands or the single value for those who only had one measurement was taken forward for analyses. People who did not have either measurement or had a measurement value of '0' for any reason were excluded ($n = 1360$). The relative grip strength (absolute grip strength divided by body mass or size measures) better reflects overall physical fitness than absolute grip strength[66] and is a stronger predictor of cardiometabolic risk[67,68]. For this study, relative maximum handgrip strength in UK Biobank was derived as the absolute maximum grip strength divided by fat-free mass measured by bioelectrical impedance analysis (Tanita BC418MA). Another 7926 participants were excluded due to missing fat-free mass measurement. A total of 478,624 individuals were included in the GWAS for fat-free mass weighted handgrip strength. The linear mixed model was adjusted for age at recruitment, sex, genotyping array and first 10 principal components.

All genetic corrections were tested after removing SNPs with MAF < 0.01, imputation quality info <0.4, located in the MHC region and with a $z$-score (beta/s.e.) >8.9 from the GWAS results. A Bonferroni corrected $p$-value was used as the significance threshold for genetic correlations ($p = 0.05$/number of pairs of traits tested).

We also examined the association between genetically predicted fitness and observed physical activity, as measured by wrist accelerometry in a subsample of 71k unrelated White European participants with at least 3 days of valid data as previously described[69]. We analysed two behavioural outcomes; total volume of movement and time spent in at least moderate intensity activity (defined as time spent above a movement-related acceleration of 150 mg)[70,71].

## Biological insights

**Fitness and proteomics.** We examined association between the 160-SNP fitness genetic risk score for cardiorespiratory fitness and protein targets assessed by the aptamer-based technology (SomaScan©) in 10,707 individuals from the Fenland study using linear regression models. The model was constructed using aptamers as outcome, weighted genetic risk score of fitness as the exposure, and included age, sex, test site (a proxy measure for sample handling which can influence the SomaLogic measures), genotyping array and the first 10 genetic principal components as covariates.

**Candidate variants and genes associated with fitness.** We used Functional Mapping and Annotation of Genome-Wide Association Studies (FUMA)[72] web-based 'SNP2GENE' function to search for candidate variants to select genome-wide significant independent signals ($r^2 < 0.6$; $p < 5 \times 10^{-8}$; default setting) and all the variants that were in LD with the identified independent signals ($r^2 \geq 0.6$; $p < 10^{-5}$, UKBB v2 Random 10 K White British cohort as the reference panel). These candidate variants were then annotated using the Ensembl Variant Effect Predictor (VEP)[73] for functional consequences, among which the missense variants were annotated using sorting intolerant from tolerant (SIFT) score and polymorphism phenotyping (PolyPhen) for predicted deleterious effects. As implemented in VEP, LoFtool was used to assess the genic intolerance and consequent susceptibility to diseases based on the ratio of Loss-of-function (LoF) to synonymous mutations, and CADD scale score was also evaluated for relative pathogenicity. Finally, we searched GWASCatalog with the candidate variants for previously associated phenotypes ($p < 5 \times 10^{-8}$).

Subsequently, to identify potentially relevant genes, the previously identified candidate variants were mapped to protein-coding genes (1) located in or in proximity to (within 10 kb) significant fitness-associated loci defined by the independent signals, (2) significantly associated with expression quantitative trait loci (eQTLs) among the candidate variants (FDR ≤ 0.05) based on a range of database (GTEx v8, BRAINEAC, BIOS QTL, Blood eQTL, MuTHER, eQTLGen, PsychENCODE, DICE, scRNA eQTLs) or (3) showed significant chromatin interaction activities with candidate variants in the CRF-associated loci (using the FUMA default options on all the four built-in chromatin database, including Hi-C of 21 tissue and cell types from GSE87112, Hi-C loops from Giusti-Rodriguez et al. 2019, Hi-C based data from PsychENCODE and Enhancer-Promoter correlations from FANTOM5)[72].

To find genes that are likely to be involved with biological mechanisms underlying fitness, we also used MAGMA and DEPICT (see details below). We combined the lists of prioritised genes from these two methods with previously mapped genes by FUMA to constitute a prioritised list of genes, which was then used as the input for FUMA 'GENE2FUNC' analyses. A systematic search for genes linked with monogenic and Mendelian diseases was completed using OMIM database (www.ncbi.nlm.nih.gov/omim), drug-gene pairs were identified from GeneCards v5.0 database (contains 20,916 protein-coding genes; www.genecards.org) and T2D Knowledge Portal (contains 135 datasets and 261 traits) (www.type2diabetesgenetics.org) for any type 2 diabetes related functions.

**Gene and gene-set based enrichment analyses.** A variety of gene-based and gene-set enrichment analyses have been developed to extrapolate relevant functional information to gain biological insights from genetic association studies. Some approaches apply to prioritised genes mapped to associated genetic loci of genome-wide or suggestive significance with the trait of interest[32,74], whilst others utilise the full genome-wide association summary statistics[75,76]. However, no defined formula exists in guidance of which approach or approaches are the most appropriate and often each approach yield somewhat variable results. Therefore, in this study, we utilised LDSC-SEG, DEPICT, MAGMA, FUMA DEG and MetaXcan to obtain a more well-rounded picture.

**LDSC-SEG.** We used the stratified LDSC applied to specifically expressed genes (LDSC-SEG) method[33] to the GWAS summary statistics in order to identify CRF-relevant tissue and cell types. The 53 tissue and cell type-specific expression data from GTEx v7 and 152 from Franke Lab were analysed jointly, and tissue and cell type-specific chromatin-based annotations from peaks for 6 epigenetic marks, including 93 labels from Encyclopedia of DNA Elements (ENCODE) EN-TEx and 396 from Roadmap Epigenomics database were used respectively for validation. False Discovery Rate (FDR) ≤ 0.05 was used as the significance threshold for enrichment.

**DEPICT.** DEPICT (Data-driven Expression Prioritized Integration for Complex Traits)[32] was developed by Pers et al., who utilised information on co-regulation of genes based on expression data and existing annotated gene sets to predict gene functions and generate 'reconstituted' gene sets. Each gene is functionally characterised by its membership probabilities across all reconstituted gene sets, and each reconstituted gene sets contain genes with various membership probabilities across the genome.

We applied DEPICT (version 1 rel194) on independent genetic variants pre-clumped using distance clumping (>1 Mb) on SNPs with a significance level of $p < 10^{-5}$ in the CRF GWAS results. Associated genes were selected using positional mapping, either within or overlap with the CRF-associated loci (LD lock defined by variants have $r^2 > 0.5$ with the independent variants) or closest to the independent variants if no genes were positionally mapped. Genes were prioritised if they

share predicted functions with genes from the other fitness-associated loci more often than expected by chance. We also assessed whether any of the reconstituted gene sets of specific biological pathways and cellular processes were significantly enriched for CRF-associated genes. To gain more biological insights, we also utilised DEPICT to identify tissue or cell types where the expressions of fitness-associated genes were enriched based on the data from a set 37,427 expression microarray samples from a total of 209 human tissues and cell types.

**MAGMA.** MAGMA (Multi-marker Analysis of GenoMic Annotation) utilised the full genome-wide association summary statistics and data from the GTEx project to identify gene-sets that are enriched based on 5500 'Curated gene sets' and 9,996 Gene Ontology (GO) terms obtained from MsigDB v7.0[75]. Rather than using a permutation-based method, MAGMA's gene analysis applies a multiple regression principal component approach, which takes into account of the LD structure between the SNPs, to compute gene-level $p$-values using an F-test. When only using summary-level data as input, the gene-analysis test was conducted using the mean of the χ2 statistics for the SNPs in the gene.

We applied MAGMA v1.6 embedded in FUMA (SNP-wide mean model) on GWAS results of CRF to perform gene-based and gene-set analyses, where SNPs were assigned to all the protein-coding genes from Ensemble build 92. The genes that were genome-wide significant after Bonferroni-correction ($p < 0.05/19,208$) were added the prioritised gene list mentioned previously. The competitive gene-set analysis was then performed using the gene analysis results, to test whether the genes in a gene-set are more strongly associated with CRF than genes not in the gene set using a one-sided two-sample $t$-test. We also used MAGMA to identify tissue-specific enrichments in 30 general tissue types and 54 specific tissue type expression data from GTEx v8.

Furthermore, in contrast with the genome-wide approach applied in MAGMA, we also used FUMA to interrogate tissue-specific expression using differentially expressed gene (DEG) sets. The DEG sets were pre-calculated by two-sided $t$-test for a given gene significantly more or less expressed in one tissue compared with all the other tissue types based on normalized expression levels. The group of genes with significant $p$-value after Bonferroni correction and absolute log fold change ≥0.58 were defined as a DEG set in a given tissue type. Similarly, by taking the direction of $t$-test into account, upregulated and downregulated DEG sets were defined. We tested our 140 prioritised genes against the DEG sets using hypergeometric test for any overrepresentation in DEG sets in specific tissue types. A tissue type with test $p$-value ≤ 0.05 after Bonferroni correction was considered enriched.

The prioritized genes were also tested against existing pre-defined gene sets obtained MSigDB (such as hallmark gene sets, KEGG, Reactome, gene sets, computational gene sets, GO gene sets, oncogenic and immunologic signatures) and WikiPathway for over-representation of biological functions and cellular processes using hypergeometric tests (FDR ≤ 0.05).

**MetaXcan.** MetaXcan[76] infers the results of PrediXcan[77], which applies a gene-based approach to test the mediating effects of predicated gene expression levels (using models trained with eQTL data from GTEx) on phenotype, to prioritise genes with regulatory mechanisms on the phenotype using summary-level GWAS statistics instead of individual-level data. In addition, MetaXcan can also perform tissue-specific analysis using models made available through PredictDB (http://predictdb.org)[76].

In this study, we used MetaXcan[76] to identify genes whose predicted transcription levels were significantly associated with fitness in specific tissue types. Up to 25,834 genes were tested in 48 different tissue types. Bonferroni correction was applied to the MetaXcan test $p$-values (significance level $p = 0.05/248,520$ tests $= 2.01 \times 10^{-7}$) to select genes that show tissue-specific altered gene expressions.

**Reporting summary**

Further information on research design is available in the Nature Portfolio Reporting Summary linked to this article.

## Data availability

The genotypic, proteomic, metabolic and phenotypic data used in this paper from the two cohort studies are available under restricted access for ethical and regulatory reasons. The results from UK Biobank presented here use applications 408, 12871 and 44448. Access to the UK Biobank data is open to all approved health researchers (http://www.ukbiobank.ac.uk/). The Fenland study data can be requested by bona fide researchers for specified scientific purposes via the study website (https://www.mrc-epid.cam.ac.uk/research/studies/fenland/information-for-researchers). Data will either be shared through an institutional data sharing agreement or arrangements will be made for analyses to be conducted remotely without the need for data transfer Source data are provided with this paper.

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

## Acknowledgements

We are grateful to the participants in the UK Biobank and Fenland studies for their time and effort. Proteomic measurements in the Fenland Study were supported and governed by a collaboration agreement between the University of Cambridge and SomaLogic. L.C. was funded by Cambridge Trust. The Fenland Study (https://doi.org/10.22025/2017.10.101.00001) is supported by the UK Medical Research Council (MC_UU_00006/1). All authors were supported by the UK Medical Research Council [MC_UU_12015/1, MC_UU_12015/2, MC_UU_12015/3, MC_UU_00006/1, MC_UU_00006/2, MC_UU_00006/4]. E.W. is now an employee of Astra-Zeneca. L.C. is now an employee of Novo Nordisk Ltd. J.R.B.P. is now an employee of Adrestia Therapeutics Ltd.

## Author contributions

L.C., N.J.W., J.R.B.P. and S.B. designed the analysis and drafted the manuscript. L.C., T.G., F.R.D. and E.W. undertook the analyses. N.D.K., E.W. and C.L. led on the analysis of the proteomic data and the diabetes endpoint definition in UK Biobank. N.J.W. and S.B. are principal investigators of the Fenland Study. All authors contributed to the interpretation of the results and critically reviewed and redrafted the manuscript.

## Competing interests

The authors declare no competing interests.
