## [Peer Review File · Nature Communications]

Causal associations between cardiorespiratory fitness and type 2 diabetesREVIEWER COMMENTS

Reviewer #1 (Remarks to the Author):

This paper uses genetics to assess causal relationships between cardiorespiratory fitness and diabetes. This paper is well written and represents a good use of the fitness data captured in UK Biobank. It provides important novel findings for the field and adds to the current evidence base highlighting the importance of fitness in health and wellbeing. The analyses are comprehensive and consider the nuances of MR. I have a few minor comments/queries below:

1. For the 160 variants used in the fitness instrumental variable did you look at how this predicted physical activity in the ~100,000 individuals with accelerometer data available? Or check a genetic correlation between fitness and PA? This would be helpful with the 14 variants as well as the 160.
2. Did you exclude variants that might potentially act via alternative pathways e.g. adiposity?
3. Clarity of figure 5 is not good, currently quite hard to read.
4. Given multiple testing the multivariable MR results are quite nominal - did you look at performing a mediation analysis to look at the % mediated?

Reviewer #2 (Remarks to the Author):

This is a very well written paper, as to be expected from this team of authors. It covers the important topic of improving our understanding of the association between cardiorespiratory fitness and type 2 diabetes. This is particularly important as cardiorespiratory fitness is a modifiable risk factor that has the potential to be altered via behavioural and/or possibly pharmacological routes. This study represents a major step forward by combining resting heart rate, ECG, genetic, and proteomic modalities from the UK Biobank and Fenland datasets.

Before recommending for publication, I have one major comment followed by a few minor comments:

=== Major comment ===

- Why didn't you construct a polygenic risk score from the UK Biobank fitness data alone?

i.e. What you would really like is to derive an instrument using fitness/ECG data alone, rather than using proxy resting heart-rate data. However, you've used heart rate data because there are only 14 instrument variables for fitness. As a result you use heart rate instrument variables, 84 (~52%) of which are weakly ($p > 0.005$) associated with fitness (Supplement Table 6). One could reasonably argue that the validation of your instrument on Fenland fitness data in Figure S4 is somewhat underwhelming as a consequence of SNPs that are not directly targeted to the trait of interest.

Instead, a polygenic risk score could help better explain your trait variation. i.e. train a polygenic risk score on the ~70k UKBB participants with fitness data, then validate on the ~12k Fenland subjects, and then perform 2 sample MR versus DIAMANTE data (which should ideally exclude UKBB Fitness participants).

==== Minor comments ====

Intro, lines 70-71 - Please be more specific on the problems/biases of the Hanscombe paper (reference 13).

Results, lines 130-132 - please provide a supplement table/figure to backup claim that (40 instrument variables) MR on fitness was compatible with a causal relationship

Results, line 138 - Please make sure to upload supplement Figure 3 - I couldn't find this anywhere

Methods, line 409 - fix typo 'ICD10 codes use**d** to identify likely type 2 diabetes...'

Methods line 419 - please state exact categories used for meat intake, oily fish intake, fruit/veg intake, etc.

Methods ,line 581 - fix spelling/typo 'grip streng**ht**'

Methods, line 644 - fix broken reference to LDSC-SEG (no reference 181 exists)

Methods, line 683 - for consistency, change 'I' to 'we'

Figure 2 - Adjust y-axis scaling, going from 0-2 should be fine and would result in a more visually pleasing plot

Reviewer #3 (Remarks to the Author):

Cai et al. examined the genetic determinants of cardiorespiratory fitness in 450k European-ancestry individuals in UK Biobank, by leveraging the genetic overlap between fitness measured by an exercise test and resting heart rate. The authors identified 160 fitness-associated loci which were validated in an independent cohort, the Fenland study. Gene-based analyses prioritized candidate genes, such as CACNA1C, SCN10A, MYH11, MYH6, that are enriched in biological processes related to cardiac muscle development and muscle contractility. In a Mendelian Randomisation framework, the authors demonstrate that higher genetically predicted fitness is causally associated with lower risk of type 2 diabetes independent of adiposity. Integration with proteomic data identified N-terminal pro B-type natriuretic peptide, hepatocyte growth factor-like protein and sex hormone-binding globulin as potential mediators of this relationship.

The study is in general well-conducted with manuscript well-written. Much effort was spent on the identification and optimization of fitness-related genetic instruments. My biggest concern was the overlapping of individuals for the GWAS of fitness and resting heart rate in UKB, these phenotypes are derived from the same individuals, for which reason a high and perhaps inflated correlation could be expected.

The overall flow can be improved, there are a few places I feel can be connected better, or simply, explained better.

In the results section, GWAS of fitness, the authors could mention a total of 14 SNPs were identified (right in that paragraph).

Would the reverse genetic correlation between fitness and RHR ($r_g = -0.68$) be influenced by sample overlap? How about the estimates in the secondary analyses including relevant traits such as lung function, handgrip strength and haemoglobin, would these estimates be influenced by sample overlap?

The authors looked up the 14 fitness-SNPs and the 40 RHR-SNPs in each other's GWAS respectively, how about performing a correlation analysis on the effect sizes of those genetic variants, in addition to the genome-wide fashion (LDSC).

An MR was performed to test the causal association between RHR and fitness, how about the reverse

direction? Would the relationship still hold if do the same for fitness (as an exposure) and RHR (as an outcome)?

When RHR GWAS was enlarged to the whole UKB (in addition to the subsample who were included in the exercise-based fitness GWAS), were these 426 identified loci looked up in the fitness GWAS? Why not?

In MVMR with intermediate traits, would be good to complement with an estimate of proportion mediated (PM value), so that the authors understand the extent to which the intermediate traits explained for the relationship.

Downstream analysis are fine, despite many alternative approaches to reach the same goal, e.g., TWAS analysis to identify genes and tissue specificity.

1. REVIEWER 1 COMMENTS

This paper uses genetics to assess causal relationships between cardiorespiratory fitness and diabetes. This paper is well written and represents a good use of the fitness data captured in UK Biobank. It provides important novel findings for the field and adds to the current evidence base highlighting the importance of fitness in health and wellbeing. The analyses are comprehensive and consider the nuances of MR. I have a few minor comments/queries below:

- 1.1 For the 160 variants used in the fitness instrumental variable did you look at how this predicted physical activity in the ~100,000 individuals with accelerometer data available? Or check a genetic correlation between fitness and PA? This would be helpful with the 14 variants as well as the 160.

Physical activity is a different construct to fitness but they are correlated and some of the variation in fitness between people is explained by engagement in physical activity, particularly that of moderate and vigorous intensity. In the revised paper, we have added an analysis of the association with accelerometry-assessed physical activity in UK Biobank.

The 160-SNP fitness-GRS was positively associated with overall volume of physical activity as well as time spent in moderate-to-vigorous activity in UK Biobank. We have added a brief comment to highlight this observation.

- 1.2 Did you exclude variants that might potentially act via alternative pathways e.g. adiposity?

- 1) Our GWAS was performed on fitness adjusted for BMI, which lowers the probability of including obesity SNPs in the fitness score. We conducted a genetic correlation analysis between fitness and BMI, and this association was not statistically significant.
- 2) We conducted multi-variable MR to adjust for the potential effect of BMI. The results showed that the potential causal association between fitness and T2D was only weakly mediated by obesity (12.2%).

- 1.3 Clarity of figure 5 is not good, currently quite hard to read.

We have improved the quality of the figure, and have also annotated the four significantly enriched tissues in cardiovascular system.

- 1.4 Given multiple testing the multivariable MR results are quite nominal - did you look at performing a mediation analysis to look at the % mediated?

We have changed the expression of the exposure units in this analysis to increase the resolution of reported results. We have also added a column in figure 3 to more clearly denote the percentage of mediation.

2. REVIEWER 2 COMMENTS

This is a very well written paper, as to be expected from this team of authors. It covers the important topic of improving our understanding of the association between cardiorespiratory fitness and type 2 diabetes. This is particularly important as cardiorespiratory fitness is a modifiable risk factor that has the potential to be altered via behavioural and/or possibly pharmacological routes. This study represents a major step forward by combining resting heart rate, ECG, genetic, and proteomic modalities from the UK Biobank and Fenland datasets.

Before recommending for publication, I have one major comment followed by a few minor comments:

2.1 Why didn't you construct a polygenic risk score from the UK Biobank fitness data alone?

i.e. What you would really like is to derive an instrument using fitness/ECG data alone, rather than using proxy resting heart-rate data. However, you've used heart rate data because there are only 14 instrument variables for fitness. As a result you use heart rate instrument variables, 84 (~52%) of which are weakly ($p > 0.005$) associated with fitness (Supplement Table 6). One could reasonably argue that the validation of your instrument on Fenland fitness data in Figure S4 is somewhat underwhelming as a consequence of SNPs that are not directly targeted to the trait of interest.

Instead, a polygenic risk score could help better explain your trait variation. i.e. train a polygenic risk score on the ~70k UKBB participants with fitness data, then validate on the ~12k Fenland subjects, and then perform 2 sample MR versus DIAMANTE data (which should ideally exclude UKBB Fitness participants).

We agree with the reviewer that a UKBB derived polygenic risk score - i.e a genetic instrument comprised of all genetic variants in the genome – might explain more of the phenotypic variation in our validation cohort. However, our intention was to use strongly associated genetic variants for the purpose of investigating causal inference using Mendelian Randomization approaches. It is considered best practice to only include genome-wide significant SNPs in such analyses. Thus we would prefer to report the Mendelian Randomization analyses as currently implemented.

2.2 Intro, lines 70-71 - Please be more specific on the problems/biases of the Hanscombe paper (reference 13).

In Gonzales et al. (2021), we show that the method used by Hanscombe et al. (2021) to estimate fitness (the naïve regression method) results in overestimation bias. This bias is caused by the fitness test risk stratification process, which was designed to make the test easier for participants who are at high risk of cardiovascular disease. Thus, while Hanscombe's findings likely reflect some aspect of fitness, they also reflect the influence of this risk stratification process.

- 2.3 Results, lines 130-132 - please provide a supplement table/figure to backup claim that (40 instrument variables) MR on fitness was compatible with a causal relationship

We apologise for omitting supplementary figure 3 in our original submission. We originally included a separate GWAS on resting HR (RHR) in the subsample with exercise-assessed fitness to match power (identifying 40 RHR SNPs) but we have now omitted this analysis and now only include the RHR GWAS among the full UKBB cohort (identifying 427 distinct genome-wide significant RHR-associated variants) and the association between the resulting RHR instrument and observed fitness. This causal relationship is shown in Supplementary Figure 3(b).

- 2.4 Results, line 138 - Please make sure to upload supplement Figure 3 - I couldn't find this anywhere

We apologise for the omission. We have added 'Supplementary Figure 3' to 'Supplementary Materials'.

- 2.5 Methods, line 409 - fix typo 'ICD10 codes use**d** to identify likely type 2 diabetes...'

We have fixed the typo.

- 2.6 Methods line 419 - please state exact categories used for meat intake, oily fish intake, fruit/beg intake, etc.

We have provided additional information about continuous and categorical variables used in observational analyses.

- 2.7 Methods ,line 581 - fix spelling/typo 'grip streng**ht**'

We have fixed the typo.

- 2.8 Methods, line 644 - fix broken reference to LDSC-SEG (no reference 181 exists)

We have fixed the broken reference.

- 2.9 Methods, line 683 - for consistency, change 'I' to 'we'

'I' has been changed to 'we'.

- 2.10 Figure 2 - Adjust y-axis scaling, going from 0-2 should be fine and would results in a more visually pleasing plot

We have adjusted the y-axis scaling (note, log-scale so no zero).

3. REVIEWER 3 COMMENTS

- 3.1 Cai et al. examined the genetic determinants of cardiorespiratory fitness in 450k European-ancestry individuals in UK Biobank, by leveraging the genetic overlap between fitness measured by an exercise test and resting heart rate. The authors identified 160 fitness-associated loci which were validated in an independent cohort, the Fenland study. Gene-based analyses prioritized candidate genes, such as CACNA1C, SCN10A, MYH11, MYH6, that are enriched in biological processes related to cardiac muscle development and muscle contractility. In a Mendelian Randomisation framework, the authors demonstrate that higher genetically predicted fitness is causally associated with lower risk of type 2 diabetes independent of adiposity. Integration with proteomic data identified N-terminal pro B-type natriuretic peptide, hepatocyte growth factor-like protein and sex hormone-binding globulin as potential mediators of this relationship.

The study is in general well-conducted with manuscript well-written. Much efforts were spent on the identification and optimization of fitness-related genetic instruments. My biggest concern was the overlapping of individuals for the GWAS of fitness and resting heart rate in UKB, these phenotypes are derived from the same individuals, for which reason a high and perhaps inflated correlation could be expected.

The overall flow can be improved, there are a few places I feel can be connected better, or simply, explained better.

We have revised the manuscript and hope that the flow is better. As shown in our independent validation of the genetic instruments, the 160-SNP instrument enhanced with resting heart rate loci outperformed the other instruments.

- 3.2 In the results section, GWAS of fitness, the authors could mention a total of 14 SNPs were identified (right in that paragraph).

We agree and now state that 14 SNPs were identified in the GWAS of exercise test-based fitness in that paragraph.

- 3.3 Would the reverse genetic correlation between fitness and RHR ($r_g = -0.68$) be influenced by sample overlap? How about the estimates in the secondary analyses including relevant traits such as lung function, handgrip strength and haemoglobin, would these estimates be influenced by sample overlap?

We do not think that this is an issue since genetic correlation analyses conducted using LD score regression are not impacted by overlapping samples, as noted in the paper (<https://www.nature.com/articles/ng.3211>)

- 3.4 The authors looked up the 14 fitness-SNPs and the 40 RHR-SNPs in each other's GWAS respectively, how about performing a correlation analysis on the effect sizes of those genetic variants, in addition to the genome-wide fashion (LDSC).

We apologise that we accidentally omitted supplementary figure 3 in our original submission; this figure shows the results requested.

- 1) Correlations of effect sizes of the variants were calculated. See Supplementary Figure 3
- 2) We have now removed RHR GWAS among the same subgroup of participants in the fitness GWAS to make the paper flow better as the Reviewer suggested. However, we did conduct genome-wide correlation between fitness GWAS and RHR GWAS among the same subgroup of participants previously as sensitivity analyses but these results are not included in the paper. The genetic correlation was $r_g = -0.66$ ($p = 7.2E-94$), which is slightly weaker compared with using the full RHR GWAS ($r_g = -0.68$; $p = 5.4E-120$), but consistent in direction and level of statistical significance.

- 3.5 An MR was performed to test the causal association between RHR and fitness, how about the reverse direction? Would the relationship still hold if do the same for fitness (as an exposure) and RHR (as an outcome)?

The Reviewer is correct that this does hold; Supplementary Figure 3 shows this.

- 3.6 When RHR GWAS was enlarged to the whole UKB (in addition to the subsample who were included in the exercise-based fitness GWAS), were these 426 identified loci looked up in the fitness GWAS? Why not?

We have now replaced the subsample RHR GWAS with the full sample RHR GWAS and looked up the effect of the resulting 427-SNP instrument in the fitness subsample; see panel b in Supplementary Figure 3. Then, to use RHR-associated variants as instrument for fitness, we conducted a Radial MR analysis using the identified 427 RHR loci as exposure on fitness to filter out the loci that are not associated with fitness. We then tested this against other instrumental variables for fitness. The results are shown in Supplementary Table 5.

- 3.7 In MVMR with intermediate traits, would be good to complement with an estimate of proportion mediated (PM value), so that the authors understand the extent to which the intermediate traits explained for the relationship.

We have now added this in Figure 3.

- 3.8 Downstream analysis are fine, despite many alternative approaches to reach the same goal, e.g., TWAS analysis to identify genes and tissue specificity.

We agree.

REVIEWERS' COMMENTS

Reviewer #1 (Remarks to the Author):

The revised manuscript is much improved and I have no further comments.

Reviewer #2 (Remarks to the Author):

The authors have addressed all my comments. I don't feel that resting heart rate is a brilliant proxy for physical fitness, but respect the authors decision to take this approach. This work represents an exciting step forward in the field by combining resting heart rate, ECG, genetic, and proteomic modalities from the UK Biobank and Fenland datasets.

I therefore fully support this paper being published and have no further requested revisions.

Reviewer #3 (Remarks to the Author):

I have no further comment.